# Performance Evaluation of Offline Motion Preparation Approaches on the Example of a Non-Linear Kinematics

**Olaf Holowenko [1],\*** **, Lucas Drowatzky [1] and Steffen Ihlenfeldt [1,2]**

1   Institute of Mechatronic Engineering, Technische Universität Dresden, 01069 Dresden, Germany;
    Lucas.Drowatzky@TU-Dresden.de (L.D.); Steffen.Ihlenfeldt@TU-Dresden.de (S.I.)
2   Fraunhofer IWU, 01187 Dresden, Germany
\*   Correspondence: Olaf.Holowenko@TU-Dresden.de

**Abstract:** In motion control, the generation of motion profiles for non-linear kinematics is usually computationally complex. In order to minimize the workload on the machine's control system, the approach pursued is outsourcing complex calculation tasks to the offline area. In this offline motion preparation, predefined criteria have to be taken into account to guarantee process stability on the real machine. During the motion preparation, a high performance is desired, characterized by less data generated and at the same time little computing effort. The evaluation will use the example of a motion specification, which is characterized by a large amount of data compared to conventional motion specifications. Thus, the demands on performance become even higher. This paper examines the performance of different motion preparation approaches known from literature. On the one hand, selected spline-based algorithms are discussed and compared. A recursive algorithm based on monomial splines is recommended for use in the example. On the other hand, a very simple approach based on the linearization of the non-linear workspace of the mechanism is presented and applied on the algorithms. With this, the performance increased significantly again.

**Keywords:** motion control; spline algorithm; B-spline; monomial spline; performance evaluation; nonlinear kinematics

## 1. Motivation

Processing machines are used for the automated production of mass consumer goods in very high quantities. A characteristic of this type of machines is the execution of cyclical motions. From an economic point of view, a high operating speed with low downtimes is necessary.

In general, processing machines are not stand-alone machines, but consist of several linked machines and buffers [1]. If downtimes occur on a single machine, e.g., due to problems within the process, the buffers are emptied by the following machine. After solving the problem, all machines continue operating and filling up the buffers again. This causes the operation speed to vary at the individual machines.

Regardless of the operating speed used, meeting specific requirements is essential, especially with regard to product quality and process stability. Otherwise, executing the process on the machine is not possible. This will be illustrated on the example of intermittent conveying of pieced goods used in the following, Figure 1a. In this process, a servo-driven non-linear kinematics is moving a comb-shaped tool in two dimensions. The comb in turn moves a product (e.g., a chocolate bar) from one rest position to the next. As motion specification, the specially optimized motion shown in Figure 1b is used [2]. In this process, firstly the product is accelerated, detaches from the comb (I), and enters a free sliding phase (II). After moving the required distance, the comb actively stops the product (III). Using this new

type of process makes it possible to increase the maximum operating speed significantly compared to a conventional conveying process where the product is regularly not detaching from the comb. It has been shown experimentally that it is possible to increase the maximum operating speed for this example process significantly from 71 min$^{-1}$ to 300 min$^{-1}$ by +285% [3].

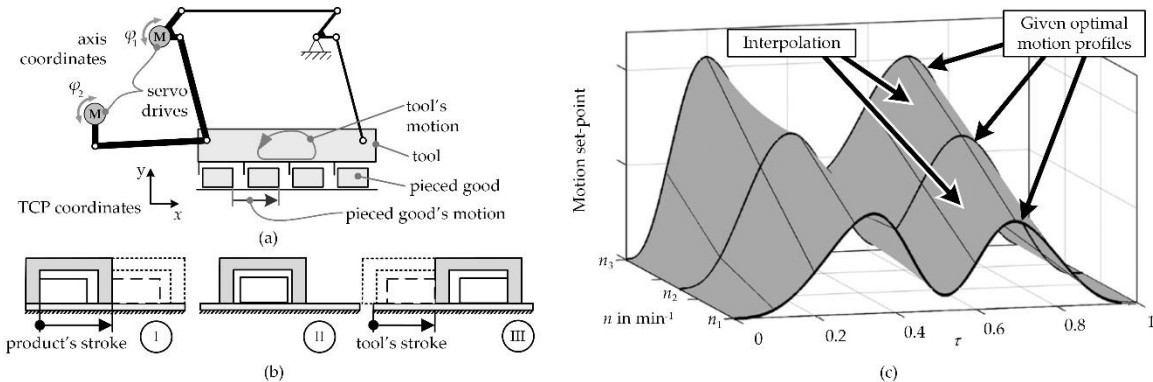

**Figure 1.** Example process. (**a**) Intermittent transport of pieced goods along a horizontal path with a servo-driven five-bar linkage. (**b**) Principles of a motion approach for the transport of pieced goods. (**c**) Characteristic map of operating speed-dependent motion profiles. Based on [2,4].

A disadvantage of the process is that with non-optimal operating speed the product may not detach from the comb at lower speeds or bounces off at higher speeds. It is then no longer positioned correctly; the process becomes unstable [3]. This process is representative for processes that vary with the operating speed, e.g., also [5].

With conventional controls, executing such unstable processes is possible only within a very small operating speed range. In previous works, a tripartite approach for controlling operation speed-dependent processes has been developed and discussed [2–6]. The first part of the approach is to optimize the motion specification for a given operating speed $n$. In doing so, effects are taken into account whose characteristics change with the operating speed, e.g., vibrations or kinetic energy. The discussion of the optimization results is not part of the work and is treated e.g., in [2,5]. Furthermore, a characteristic map of operating-speed-dependent motion profiles is created, Figure 1c. It contains several optimized motion profiles for different operating speeds, given as point lists. The motion profiles contain relevant values for the control, e.g., position as well as velocity and torque feedforward set points. The contents required for a specific process is determined during the optimization and are therefore predefined values for the motion preparation. Within the given map, the normalized cycle time $\tau$ is one dimension, the operating speed $n$ for the motion profile is the second. The optimal motion profiles are directly included in the map. If a non-optimized operating speed has to be driven (Figure 1c, gray), an interpolated value is used [4,6]. Finally, a specialized microcontroller-based control system has been developed. It allows the processing of this characteristic map of motion profiles with a very high online interpolation rate of 8 kHz. Therein, for efficient, highly dynamic motion processing, splines are used. Online aspects of the control are not part of this work; for more details see [6].

The combination of optimization, operating-speed-dependent motion profiles, and the novel control system leads to a significant increase in the achievable operating speed. At the same time, varying the operating speed over a wide range is possible. A disadvantage is that using more than one single motion specification, as in conventional systems, leads to a significantly higher amount of data needed to describe a single process. This is particularly disadvantageous against the background of real-time processing, as many controllers do not have a large data memory.

Using a very fast controller means that only a short computing time (125 µs in our example) is available online. The online motion processing of the specifications must, therefore, be as efficient as possible. To achieve this, the approach is to carry out online calculations into the offline field wherever possible. Such calculations include, in particular, computationally complex operations like kinematic

transformations. Even though computers and control systems are becoming faster and faster, other aspects speak in favor of outsourcing the calculation effort to the offline preliminary field. On the one hand, there is an increasing trend to use small, inexpensive, networked controllers with low computing capacity ("Industry 4.0", "Smart Manufacturing"). On the other hand, the idea of relieving the controller and keeping resources free for other (new) functions underlines the relevance of the approach. At the same time, the demand for individualized products increases (e.g., customized products, often in the context of "Industry 4.0" or "Smart Manufacturing"). Individualized products, in turn, require individual motion specifications that must be prepared for the control approach. This increases the amount of work needed for the offline motion preparation.

Figure 2 shows the integration of the motion preparation in the proposed workflow. In the first step, the motion specification is numerically calculated and optimized under consideration of operating-speed-dependent effects. Due to the optimization algorithm used, point lists are generated. The specifications are combined to a map of operating speed-dependent motion profiles. Subsequently, a map of splines is generated from these point lists during the motion preparation, which can be processed very efficiently later. This is done in the area of online motion processing. Previous work examined aspects of motion optimization and online motion processing, Figure 2, gray. The present work discusses offline preparation aspects.

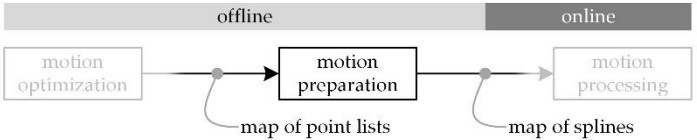

**Figure 2.** Position of the motion preparation in the workflow.

In order to reduce downtime on the machine, the motion specifications have to be prepared quickly. To be able to assess this, the work deals with the performance evaluation of different approaches for offline motion preparation known from literature. The paper is structured as follows: first, requirements on the results of the offline motion preparation and the performance evaluation are defined (Section 2). After evaluating conventional approaches for offline motion preparation, optimization potentials are deduced (Section 3). Furthermore, a very simple approach to increase the performance is discussed and evaluated (Section 4). Finally, an overview on the approaches is given (Section 5) followed by conclusions (Section 6).

## 2. Preliminary Remarks on the Offline Motion Preparation

As a basis for the offline motion preparation, motion set points (position as well as velocity and torque for feedforward control) are given for the 2D-tool center point (TCP). In the approach pursued, each of the three set points may contain several motion profiles, optimized for selected operation speeds $n$, Figure 1c. Depending on the process under consideration, up to 3 [2] or more [6] profiles may be necessary. Furthermore, the kinematic parameters of the mechanism are known, Figure 1a. All motions are within one limited workspace, which is advantageous for the optimization approach discussed here. The characteristic map of motion profiles is available as a map of point lists. The reason for this is that the optimization approach iteratively calculates single set points [2].

As the investigations in the previous work have shown, both the different motion specifications and the specifications for different processes have similar characteristics, as described above. Thus, the approach described here can be transferred to other sample processes. In the present work, for the reason of simplicity the discussion is conducted on the example of the position set points for the intermittent transport of pieced goods, Figure 1. They are also representative of other processes or motion specifications.

## 2.1. Requirements on the Preparation Result

Splines are most suitable for the online processing of the motion specifications with variable machine speeds [6]. They are widely used in the area of motion control and have several advantages, especially for the execution of operating-speed dependent motion profiles. Firstly, they scale with the variable machine speed due to their online-calculation in every interpolation cycle. Furthermore, the amount of data is significantly smaller compared to the number of given set points if tolerable deviations or limits to be fulfilled are specified.

To reduce the required data volume, splines have to approximate the optimal set points. This in turn means that not every calculated set point will meet an optimal set point exactly. Therefore, it is necessary to define an allowed tolerance respective to a target accuracy. The calculated set points have to meet those given criteria. In this work, for reasons of simplicity, the comb's positioning accuracy is used as a criterion to be fulfilled. It is calculated from the difference between the given position set point and set point calculated from the spline for both axes, assuming ideal kinematics and infinite rigidity. Real effects are not taken into account, as they do not affect the offline motion preparation. They are considered in the optimization process and are not part of this work.

An accuracy of $\varepsilon = 1$ µm is common when calculating position set points on cams [7]. This magnitude has also proven to be useful in previous studies [3,4,6]. Depending on the process under consideration, however, lower accuracies (respectively higher tolerances) may also be sufficient.

Further requirements on the result of the offline motion preparation were defined in [6]. Thus, the amount of data given to the interpolator has to be as small as possible, as online resources are limited. To ensure the short available online computation time, the characteristic map of motion profiles is to be transferred to the interpolator as a set of operating speed-dependent 1D-axis splines on the monomial basis [6]. In order to meet the optimization results, one requirement of the outgoing motion profiles is that the splines are $C^2$ continuous, particularly between the splines end and its beginning. This is necessary to guarantee a smooth, continuous motion.

## 2.2. Performance Evaluation

In this work, selected motion preparation algorithms will be evaluated for their performance. Therefore, they are applied to the process show in Figure 1. In our example, the performance depends on two parameters: the amount of data produced by the algorithm and the effort required for offline motion preparation. The required motion accuracy on the TCP has to be maintained in all cases.

For a good performance, the amount of data has to be small. The overall data quantity results from the total number of coefficients required to describe the whole motion specification for all axes. This number is the sum of all spline coefficients in the characteristic map of both axes, Figure 1c.

A low preparation effort as an equivalent to the algorithms speed is also important for a high performance. Especially against the background of individualized production and the increasingly rapid changes in the motion specifications, this point is becoming increasingly important. The offline preparation effort is determined in the same way as in [6]. The effort is measured in FLOP (floating point operations). The number of FLOPs depends on the computing operations used. For weighting, the one from [8] is used: 1 FLOP for add, subtract, multiply, and IF; 4 FLOP for divide, square root; and 8 FLOP for exponential, sine, etc. All required operations are summed up to give the computational effort, which is expressed in 1 MFLOP = $10^6$ FLOP.

To evaluate the performance of the algorithms, the 2D curve representation similar to that suggested in [9] is used. In our case, the abscissa is the amount of data calculated with the algorithm, the ordinate is the necessary number of FLOPs. The performance curve is calculated by converting the motion profiles with a specific algorithm and a technically relevant TCP motion accuracy. The resulting data quantity and calculation effort are displayed as a point in the diagram; points of one algorithm for different accuracies form together the performance curve for this algorithm. In the following, a TCP motion accuracy between 1 µm and 500 µm is used as an example to evaluate in a qualitative way how the algorithm's performance changes with differing requirements.

## 3. Conventional Offline Motion Preparation

There are different approaches for offline processing of point clouds into splines that are easy to use in the control system. In this chapter, known (*conventional*) approaches are presented and examined for their performance. Subsequently, optimization potentials are derived.

Using splines requires the examination of a large state of the art concerning splines and spline algorithms. It has been developed over decades and contains countless high-quality publications. This work is neither an introduction to splines nor a review of the state of the art concerning algorithms. For this purpose, reference is made to suitable literature, e.g., [10–13] or [14–16].

### 3.1. Relevant Types of Spline

Many works give an overview of different spline types, e.g., [10–13]. Each spline type has specific properties. Therefore, they are suitable in different ways for use in offline motion preparation.

Two types of spline have properties that make them interesting for offline motion preparation. B-splines are widely used in computer graphics and computer-aided design. With B-splines, generating continuous motions with scalable continuity is possible. Many helpful functions are available, e.g., insert or remove nodes [11]. With monomial splines, specifying derivatives at the segment boundaries can also achieve scalable continuity (Hermite interpolation). Monomial splines are much easier to implement than B-splines with the same accuracy [17]. Converting B-splines to the monomial base is easily possible by using a base transform.

Due to the properties and the requirements mentioned in Section 2.1 some widely used spline types cannot be considered for offline motion preparation. Therefore, they are not taken into account in this work. The widely used cubic splines (e.g., [13]) are not evaluated, since preliminary investigations in [18] have shown that higher order splines (e.g., quantic Hermite splines) can be used much more flexibly with about the same number of coefficients. Akima splines [19] are a type of spline often used for online interpolation in control systems. Due to their lack of $C^2$ continuity, they do not fulfill the given requirements and are not discussed as well. Furthermore, computationally more expensive approaches such as e.g., non-uniform rational basis spline (NURBS) [11] are not considered, because they do not meet the given requirements for online processing, see [6].

In this work, B-splines and monomial splines are evaluated together. Both spline types have different advantages and disadvantages, but comparable approximation properties [17,20]. On the one hand, univariate B-splines of degree 3 are used in the work to realize the required continuity. On the other hand, quintic splines are used, where $C^2$ continuity is achieved by specifying the 0th to 2nd derivative at the segment borders.

### 3.2. Relevant Preparation Algorithms

For the preparation of a given point list for one operating speed, various methods could be used, see e.g., [14–16]. However, comparative studies regarding the performance of different algorithms are rare in the literature. Therefore, statements such as "with [the introduced algorithm] LSPIA, a very large data set can be fitted efficiently" [21] are hardly verifiable. For this reason, some motion preparation algorithms using B-spline and monomial splines are evaluated for their performance. The B-spline approaches use the advantages of B-splines and are representative for other B-spline algorithms. They will be compared with two monomial spline approaches, which are expected to be much easier to calculate. All approaches promise a good performance during the offline motion preparation. The algorithms are discussed on the example of a non-linear kinematics and the process introduced in Section 1.

In the following algorithms, the function named *SplineInTolerance()* calculates the distance between the given, optimized motion set point and the point calculated from the spline. The whole spline is checked, whether it meets the given requirements. The first return value is "ok" for "in tolerance/meets accuracy" which is true, if the calculated spline meets the given requirements. The second value is

an array of indices of the segments that do not match the given accuracy. Those indices are used for inserting new knots. Depending on the given check mode, only the index for the segment with the maximum error (mode "maximum only") or all faulty segments indices are returned (mode "all"). With these two modes, two variants of the B-spline algorithms can be evaluated. If only the segment with the maximum deviation is subdivided, a small amount of data is expected with increased computational effort. If, as suggested in [22], all segments with too large deviation are subdivided, a smaller computational effort caused by less iterations are expected with a simultaneously increasing amount of data. The reason for this is that inserting a single control point also affects the neighboring segments. If several neighboring segments are faulty, more nodes than necessary may result.

*SplineSegInTolerance()* only checks one segment of the spline for compliance and can, therefore, be implemented for greater performance. *Calculate(StartEnd)Derivatives*() calculates from the given points the derivatives using central differentiation according to [23]. At the motions boundaries, the cyclical character of the motion is used. With these values a system of equations results, which can be solved. *SolveEquationSystem*() uses QR decomposition with Householder transformations to solve the equation system which has been set up in *SetupEquationSystem*().

Implementation is always important when evaluating different approaches. In order to ensure comparability, the choice of segment lengths in all algorithms is based on bisection at 50% within the considered range. Other solutions like separation at the largest error or curvature require further investigation and are not part of this work.

**bsLSQ**: the first considered approach implements [24], Figure 3, to calculate a B-spline globally. [24] uses the principle of least square approximation and thus offers a solution with minimal error for the problem. With the algorithm, derivatives can be specified at the start and end of the spline. The algorithm generates "closed curves with various degrees of smoothness" [24]. If the spline does not meet the required accuracy after calculation, additional nodes are inserted. The splitting is always performed at the given, optimized points. This avoids the knot span not containing any motion values and a singular system of equation is created. In the case of "maximum only" mode, a single node is inserted into the segment with the maximum error ($\text{bsLSQ}_{\text{max}}$); otherwise (mode "all") a single node is inserted into all segments, that exceed the accuracy ($\text{bsLSQ}_{\text{all}}$).

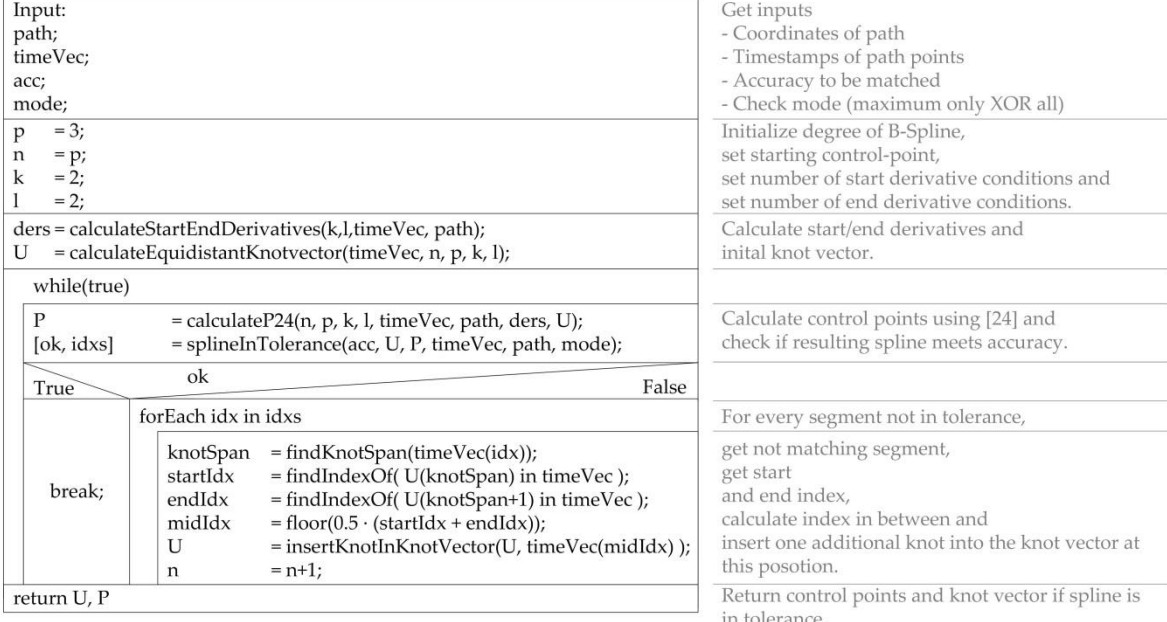

**Figure 3.** B-spline-based least square algorithm bsLSQ, using [24].

**bsITER**: in the second B-Spline approach evaluated, [21] is used in the core, Figure 4. [21] uses an iterative algorithm to approximate the numerical solution of the equation system. By extending the calculation of the first and last control points from [24], $C^2$-continuity between end of the spline and its beginning can be guaranteed. The algorithm is similar to bsLSQ; only the calculation of the solution differs. bsITER is also evaluated for the two check modes "maximum only" (bsITER$_{max}$) and "all" (bsITER$_{all}$).

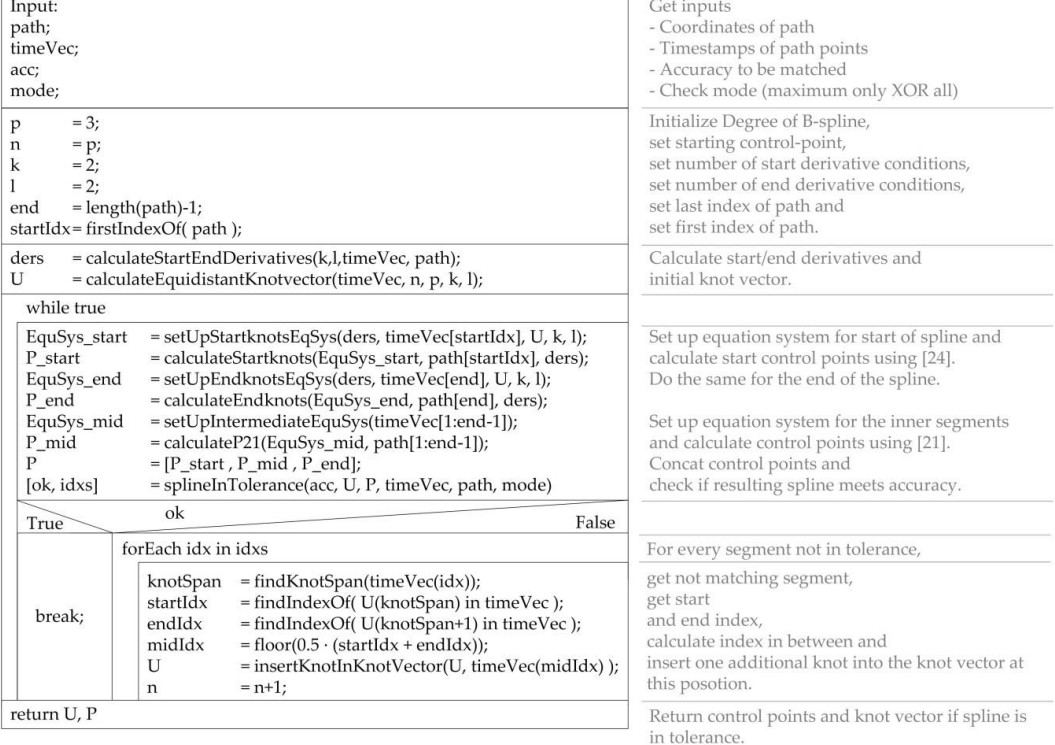

**Figure 4.** B-spline-based iterative algorithm bsITER, using [21].

**msITER**: this approach represents a very simple implementation of an iterative algorithm, Figure 5. It searches step by step for the shortest spline segment that meets the requirements. As described above, the considered segment is halved if it does not meet the requirements. The segment boundaries are thereby always set to already given, optimized points. If a segment fulfilling the requirements is found, a further run with the remaining points is performed. It is expected that the algorithm will be more performant than B-spline approaches with a similar or smaller amount of data.

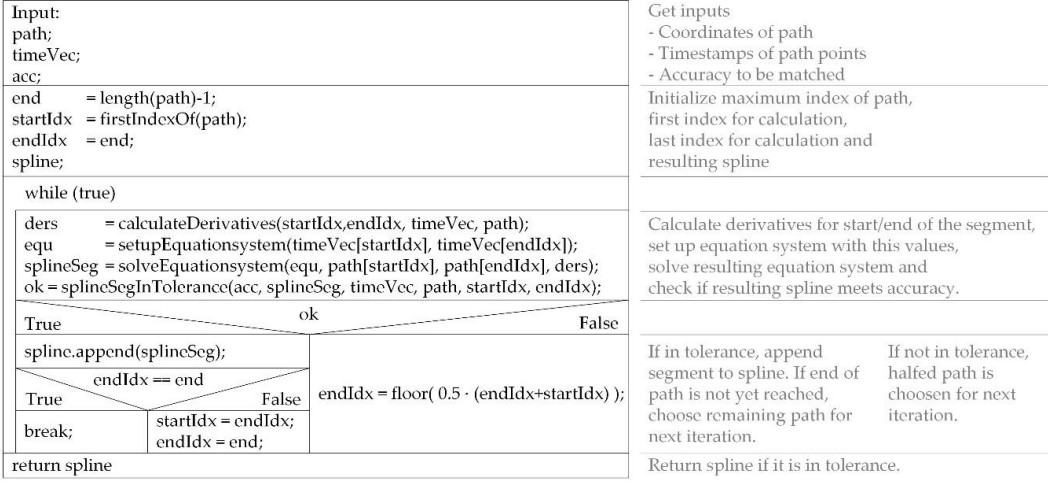

**Figure 5.** Monomial spline-based iterative algorithm msITER.

**msRECUR**: the fourth approach examined is a recursive one, Figure 6. It checks for a given part of the motion if the spline segment is as accurate as required. For this purpose, the function *calculateSplineSeg()* is initially called with the complete motion, Figure 6, below. The function checks whether a spline segment, which was calculated using the 0th to 2nd derivatives at the ends, meets the given requirements. If this check is successful, the monomial spline segment is returned as a result. Otherwise, the motion is split into two parts at 50%. Again, the segment boundaries are located at given points. Both sub-motions are passed to another instance of the function. It is expected that this algorithm will achieve a minimum calculation effort with higher data volumes than msITER. This algorithm is the only one suitable for multi-threaded implementation. However, it does not reduce the required effort, but leads to faster processing. This case is not considered here.

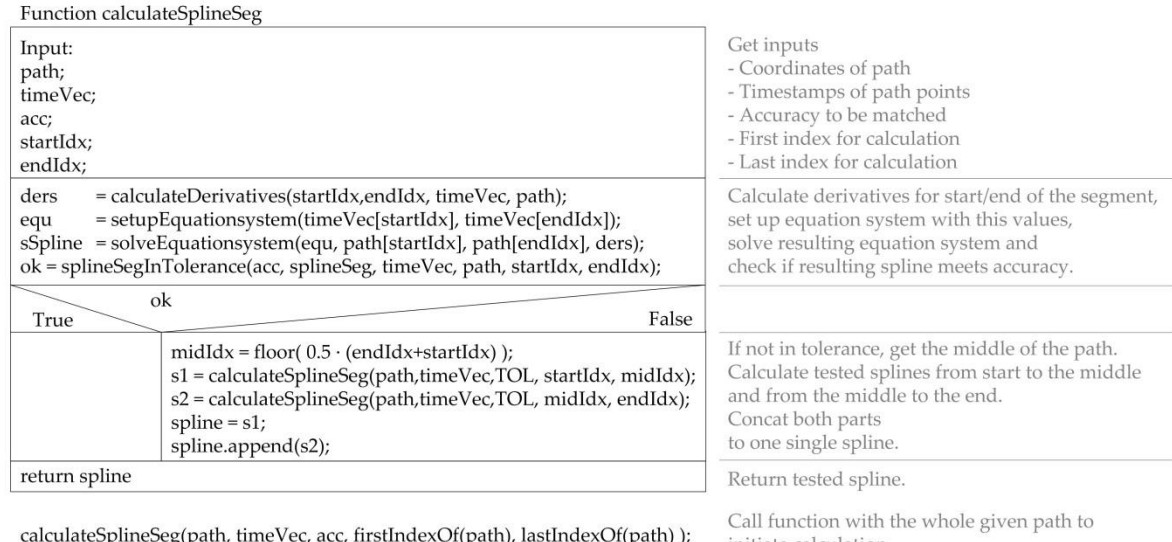

**Figure 6.** Monomial spline-based recursive algorithm msRECUR.

*3.3. Performance Evaluation*

The state of the art offers few hints on how motion preparation is performed in similar applications.

In the obvious case of offline motion preparation, both coordinates (x and y in the TCP-coordinate system; $\varphi_1$ and $\varphi_2$ in axis coordinate system, cf. Figure 1a) are considered together. In this case, the length of the spline segments in both coordinate directions is always identical compared to the normalized cycle time $\tau$. Thus, the specifications are "coupled" over the segment lengths, similar to a 2D spline. The advantage of this is that each segment can be transformed completely from the axis level to TCP level and vise-versa.

Figure 7 shows the performance of the evaluated algorithms in two 2D performance diagrams similar to [9]. As described in Section 2, the aim is to obtain a small amount of data with low calculation effort. This means, in the diagram, if a point is close to the origin, the assigned algorithm is more performant than one far away. It is then more suitable for the use in example. If two points are close to each other, it is proposed to enhance this qualitative performance evaluation method by looking at the concrete measurement results. In our example, the decision about the performance of an algorithm will be made in favor of the greater savings achieved.

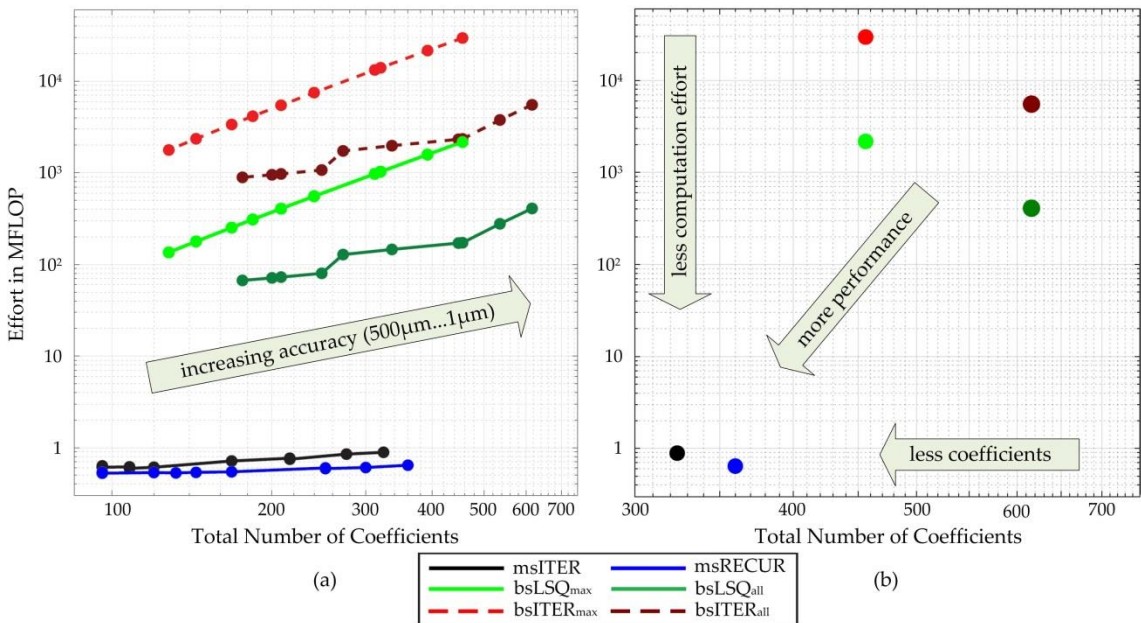

**Figure 7.** Comparison of the performance of the algorithms with coupled drive coordinates. (**a**) Trend of the performance as a function of the accuracy. (**b**) Performance for the relevant accuracy of 1 μm.

Figure 7a shows the performance curve of the algorithms over several accuracies between 1 μm and 500 μm. As expected, the performance decreases with increasing accuracy (small $\varepsilon$), Figure 7a. A closer look reveals that the performance curve of the monomial spline algorithms is approximately linear with respect to the accuracy. The gradient of the msRECUR-curve is flatter, resulting in a positive effect on the computational effort if the accuracy is raised. For the B-spline algorithms, the performance decreases following a power function with an exponent of about 2.2 (bsLSQ$_{max}$, bsITER$_{max}$) and 1.3 (bsLSQ$_{all}$, bsITER$_{all}$). This trend speaks against using B-spline algorithms in our example. From the location of the curves it can be deduced that the performance of the monomial spline algorithms is always better than the performance of the B-spline algorithms.

For an accuracy of 1 μm, Figure 7b, the B-spline based algorithms produce up to 90% more data while using up to about 46,000 times more calculation effort compared to msRECUR. Furthermore, bsITER requires about 10 times of the effort compared to bsLSQ to get sufficiently close to the exact solution. Therefore, bsITER shows the worst performance, followed by bsLSQ. This performance of the B-spline algorithms is primarily due to the very complex least square fitting in the background of those algorithms.

In the diagram Figure 7b, msITER appears closer to the origin than msRECUR, but their difference is small. In concrete numbers, for an accuracy of 1 μm msITER is 10% more efficient in terms of coefficients, while msRECUR saves 19% effort. Due to the greater savings, msRECUR is the most suitable algorithm for our example, closely followed by msITER.

*3.4. Optimization Potential*

Due to the non-linearity of the mechanism, it is assumed that the kinematic transformations make up a significant percentage of the total computational effort. It is further assumed that reducing the number of kinematic transformations while motion preparation leads to an increase in performance. This will be investigated in the following.

Figure 8a shows the amount of computational effort for all kinematic transformations in relation to the total computational effort for the algorithms. It can be clearly seen that in the B-spline algorithms this proportion is very small (<1%). This is related to the fact that the computational effort for the algorithms themselves is much higher than for msITER and msRECUR, Figure 7. With these

algorithms, the transformations require in average 61% (msITER) and 68% (msRECUR) of the total computing effort (50% and 60.7% for 1 μm accuracy). Figure 8b tries to illustrate the comparison of monomial splines (left) and B-splines (right) graphically in the diagram.

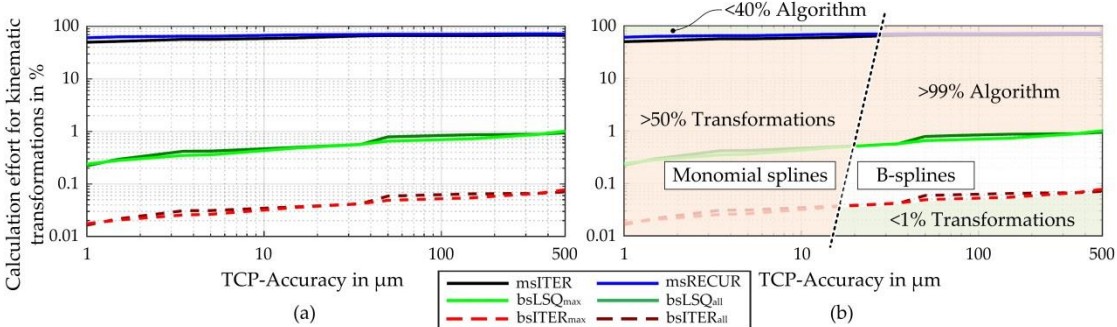

**Figure 8.** Percentage of computational effort for kinematic transformations in relation to the total computational effort. (**a**) Results for all algorithms. (**b**) Comparison of monomial splines (left) and B-splines (right).

It should be noted that in our example the kinematic transformations could be calculated with very little effort. Even with our simple kinematics, the kinematic transformations consume a very large amount of resources (up to 60.7% for the relevant accuracy of 1 μm). This lowers the efficiency of the motion preparation. Other kinematics (e.g., hexapods) are much more complex. This in turn means that the computational effort for each transformation is considerably higher (and thus also the percentage in the total computational effort). Therefore, minimizing the number of kinematic transformations seems to be reasonable for msITER and msRECUR.

Preparing coupled coordinate directions has some disadvantages. In this case, the more curved coordinate direction of the path determines the number of spline segments; less curved segments of a coordinate direction are subdivided more often than necessary. This results in more data due to the increased number of segments and thus coefficients. Therefore, another approach to optimization is the decoupling of the axes. Here, the axis splines are segmented independently. Strong curvature in a specification thereby only affects one spline, the second axis is not affected.

## 4. Simple Approach for Optimizing the Offline Motion Preparation

As discussed before, reducing the number of kinematic transformations seems to be reasonable for some algorithms to reduce the offline calculation effort, even on a simple mechanism as discussed here. For this purpose, in this work a very simple approach is discussed for its ability to reduce the offline calculation effort. The approach focusses on the question of whether linearization of the transmission in the workspace is allowed and whether this can increase the performance of the offline motion preparation. Such a linearization would have the advantage that the requirements on a motion points could be tested directly at axis level. Not a single kinematic transformation would have to be calculated while motion preparation. On the one hand, this significantly reduces the number of kinematic transformations. On the other hand, decoupling of the axes, i.e., calculating each axis individually, becomes possible.

### 4.1. General Approach

The basic idea is to convert the given accuracy on the TCP to an approximate accuracy at axis level where the splines are generated. Due to the non-linear transmission of the mechanism, this is not possible without problems. Figure 9 illustrates the non-linear effect using the example of the distortion of a circle on the TCP to an ellipse at axis level. Around the set point $p$ on the motion specification (blue, left), the given accuracy $\varepsilon = 1$ μm is shown as a black circle. All motion points that are located

within the black circle are closer to $p$ than required and, therefore, valid set points. After transformation into axis coordinates, the nominal value $\varphi$ results from $p$ (blue, right). The transformed but distorted circle is shown in black. Thereby, the distortion depends on the position of $p$ and varies over the workspace. If a calculated set point is located within the black tolerance border, it is also valid.

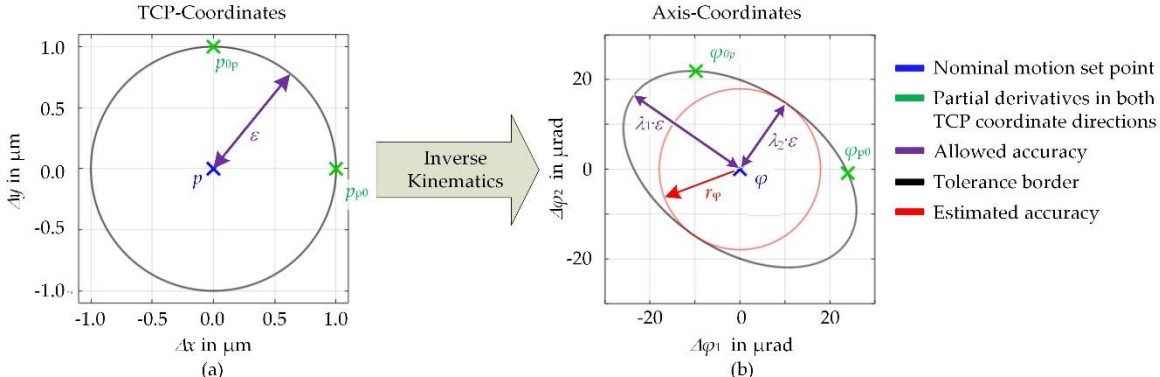

**Figure 9.** Distortion due to nonlinearity. (**a**) Tolerance circle in tool center point (TCP)-coordinate system, (**b**) distorted circle in axis coordinates.

As an approach to linearization, the inscribed circle in the ellipse with the radius $r_\varphi$ (Figure 9, red) is to be used instead of the whole ellipse. Points inside this circle are always completely within the required tolerance. Thus, the distance of a calculated spline point to the optimal set point $\varphi$ can be checked easily for compliance with the requirements at TCP level without a transformation.

$r_\varphi$ can be estimated by using the singular value decomposition (SVD) of the Jacobian matrix $J$ at point $p$ known from robotics, e.g., [25]. The Jacobian consists of the partial derivatives in the two directions of motion at $p$. In practice, it can be approximated by numerical differentiation. When using forward differences, the calculation of at least 3 transformations is necessary ($p$, $p_{p0} = p + [\varepsilon, 0]$, $p_{0p} = p + [0, \varepsilon]$, Figure 9, green). This one-sided difference represents a simplification and saves one kinematic transformation compared to the bilinear difference. For our example, the closed solution of the SVD from [26] is used for calculation with:

$$a = \left(\varphi_{p0.1} - \varphi_1\right)/\varepsilon \tag{1}$$

$$b = \left(\varphi_{0p.1} - \varphi_1\right)/\varepsilon \tag{2}$$

$$c = \left(\varphi_{p0.2} - \varphi_2\right)/\varepsilon \tag{3}$$

and

$$d = \left(\varphi_{0p.2} - \varphi_2\right)/\varepsilon \tag{4}$$

$\lambda_1$ and $\lambda_2$ are calculated by [26].

$$\lambda_1 = \frac{\sqrt{a^2 + b^2 + c^2 + d^2 + 2\cdot(ad - bc)} + \sqrt{a^2 + b^2 + c^2 + d^2 - 2\cdot(ad - bc)}}{2} \tag{5}$$

$$\lambda_2 = \frac{\left|\sqrt{a^2 + b^2 + c^2 + d^2 + 2\cdot(ad - bc)} - \sqrt{a^2 + b^2 + c^2 + d^2 - 2\cdot(ad - bc)}\right|}{2} \tag{6}$$

Therein, $\lambda_1$ corresponds to the major semi-axis and $\lambda_2$ to the minor semi-axis of the ellipse [27]. $\lambda_1$ and $\lambda_2$ have the unit rad/m and represent a transmission factor. As can be seen in Figure 9, the diameter of the circle $\varepsilon$ on the TCP is distorted to $\lambda_1$ and $\lambda_2$ on axis level. In the minimal case, $\varepsilon$ is

distorted to $r_\varphi$ by $\lambda_2$. Since for our application the radius of the inscribed circle is of interest, $r_\varphi$ has to be calculated from $\varepsilon$ and the smaller $\lambda_2$ by:

$$r_\varphi = \lambda_2 \cdot \varepsilon \tag{7}$$

This consideration requires a coupled calculation of the two axes, since both axis coordinates are included in the calculation of the distance. In the uncoupled case, $r_\varphi$ should be distributed to both axes. The circular equation:

$$r_\varphi^2 = r_{\varphi.1}^2 + r_{\varphi.2}^2 \tag{8}$$

could be used to split $r_\varphi$ to both axis coordinates. If the proportions are evenly distributed on both axes ($r_{\varphi.ax} = r_{\varphi.1} = r_{\varphi.2}$), $r_{\varphi.ax}$ results in:

$$r_{\varphi.\mathrm{ax}} = \frac{r_\varphi}{\sqrt{2}} \tag{9}$$

This allows the calculation to take place on the single axis, and decoupling of the axes is possible.

### 4.2. Workspace Evaluation

The analysis of the workspace is the basis for the evaluation of the applicability of the approach to a real manipulator [28]. It can be carried out during the design or commissioning of the machine, so it only has to be done once. This is particularly useful in our case, since a single process description contains several motion specifications that all have to be prepared.

The following diagrams show the entire TCP workspace on the left and a realistic TCP workspace for the process on the right. In our example, the TCP workspace is 10 mm larger in all directions than the given motion path.

#### 4.2.1. $\lambda_2$ in the Workspace

Figure 10 shows $\lambda_2$ over the workspace of the mechanism; the values are limited from 7 rad/m to 9 rad/m. This diagram can be used to estimate whether linearization is appropriate. The more constant $\lambda_2$ in the area of the motion is, the more likely linearization is to be used effectively. In our example, $\lambda_2$ varies on the given path between 7.07 rad/m and 7.52 rad/m. The maximum value is thus about 6.4% above the minimum value. In the realistic workspace, $\lambda_2$ varies between $\lambda_{2.\min} = 7.02$ rad/m and 8.12 rad/m, the minimum value is 14% below the maximum.

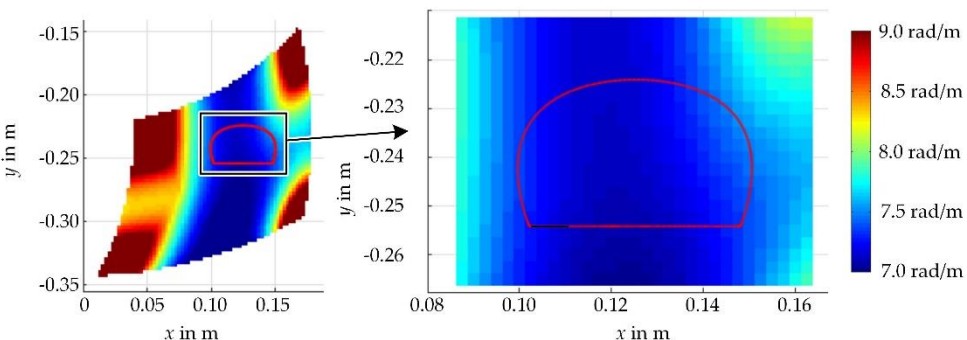

**Figure 10.** Transmission $\lambda_2$ over the workspace of the manipulator.

Considering these values, it is expected that a linearization by using $\lambda_{2.\min}$ for the whole workspace can save computing time while producing a reasonably larger amount of data. Enlarging the workspace would lead to worse results, especially at the borders of the workspace. A favorable placement of the motion path may increase the accuracy and improve the ability to linearize. It is expected that using linearization will result in a spline that is up to 14% more accurate than necessary. This would raise the number of coefficients and lower the performance of the algorithms.

### 4.2.2. Distortion of the Workspace

The ratio $d = \lambda_2/\lambda_1$ is described in [28] as inverse condition number measurement. It allows a prediction about the deformation of the circle. The ratio $d = 1$ means that the circle on the TCP is also mapped as a circle at axis level. The distortion is then very small, which has a positive effect on the quality of the transmission [25]. Figure 11 shows $d$ over the workspace, the values are limited from 1 to 2.

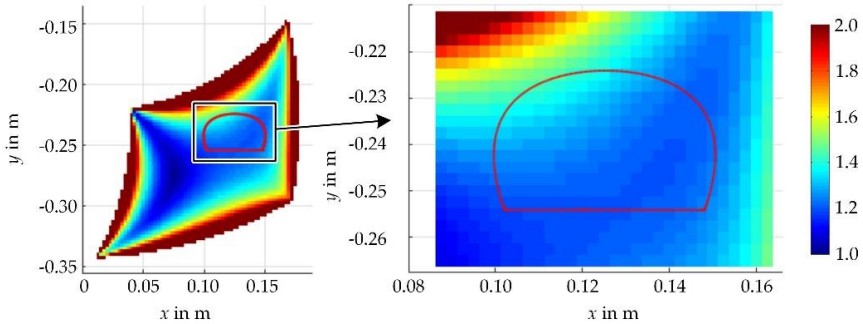

**Figure 11.** Distortion $\lambda_2/\lambda_1$ in the workspace of the manipulator.

With a ratio $d = 1$, every valid point on the TCP is also checked as valid at axis level. With a ratio of 2, the inscribed circle has only half the area of the ellipse. Then about 50% of all usable points are classified as good. The motion considered here runs between 1.18 and 1.44. This means that with linearization, at most between 69% and 85% of all usable points are recognized as good. How this affects the amount of data will be shown on the example.

The smaller $d$, the more points are correctly classified. As a result, longer spline segments will be calculated. Thus, a path in the area with a small $d$ (blue) can be mapped with fewer spline segments and less data than one in the area with large $d$. The positioning of the path in the workspace thereby also has an influence on the amount of data that is generated.

### 4.2.3. Error Estimation

Besides the linearization itself, several simplifications have been made so far. This concerns on the one hand the calculation of the Jacobian matrix. Here one-sided differences are used instead of central differences to save 25% of the transformations. This error is estimated to be negligibly small.

It is well known that with a smaller step size the Jacobian matrix becomes more accurate when differentiating, but if the steps are too small then numerical issues can cause problems. To estimate the magnitude of this, the ratio between $\lambda_2$ calculated with large $\varepsilon$ ($\varepsilon = 2$ mm) and small $\varepsilon$ ($\varepsilon = 0.1$ μm) is shown in Figure 12. From the result a maximum error of about 0.5% in the realistic workspace is estimated.

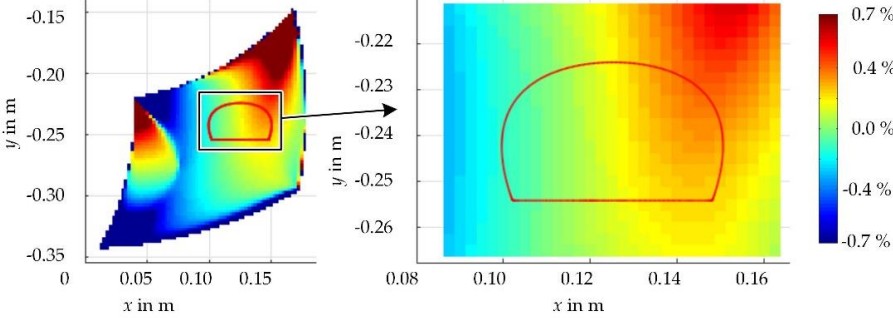

**Figure 12.** Error when using large $\varepsilon$ instead of small $\varepsilon$ for calculating the Jacobian in the manipulator's workspace.

Further errors result from the calculation of $\lambda_2$ in the workspace, Figure 10. If too few points (e.g., only the four corners of the workspace) are calculated, the resolution is too low, a minimum cannot be found with sufficient certainty. Depending on the selected grid, the global minimum may be located between the grid points. In order to stay safely below, its value is to be estimated. Therefore, the points (Figure 13, light red and red) surrounding the minimum (black) are used. Their maximum (red) should be selected and extrapolated beyond the minimum point (black). The value of the resulting estimated point (blue) is to be used for calculating an estimated minimum error. The finer the grid, the more accurately the minimum can be calculated; but this is associated with increased additional calculation effort. In our example, a 2.5 mm grid in $x$ and $y$ direction was used (about 30 points in $x$ direction). The error results in being 0.23%.

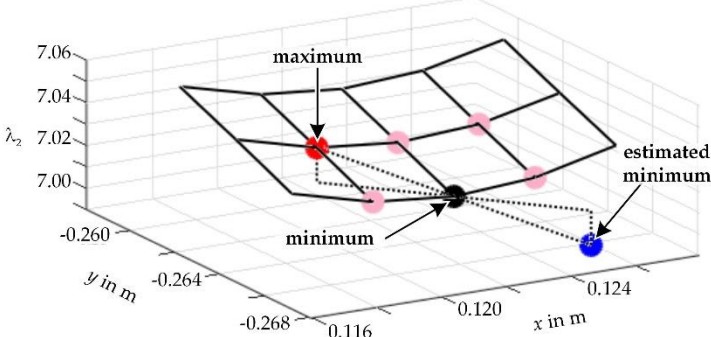

**Figure 13.** Estimation of a minimum $\lambda_2$ (blue) by extrapolation between the minimum (black) and the surrounding maximum (red).

Splitting $r_\varphi$ over the two axes in Equation (8) does not cause errors. Using this approach, only a square inside the inscribed circle in the ellipse is considered. Due to the smaller area tested, it is expected that this will result in at most 64% of all suitable set points being declared as good. In this case, the number of coefficients for a given motion accuracy will increase. In addition, the spline will be closer to the optimized target set point, the accuracy of the spline will be higher than required.

Overall, the simplification results in a total estimated error of about 1%. The following calculations are therefore made with $\lambda_{2.\text{corr}} = 0.99 \cdot \lambda_{2.\text{min}} = 6.95$ rad/m. At axis level, $\lambda_{2.\text{corr.ax}} = 1/\sqrt{2} \cdot \lambda_{2.\text{corr}} = 4.91$ rad/m will be used. With a known TCP accuracy, an axis accuracy can be calculated by Equation (7).

### 4.3. Performance Evaluation

With this approach and the previous knowledge about the workspace of the mechanism, there are many possibilities for optimizing the offline motion preparation. For the sake of simplicity, a single scenario will be considered, which could be extended to other applications in the following studies.

In this scenario, two basic definitions are made. On the one hand, a constant transmission $\lambda_2$ over the entire workspace is estimated. In doing so, all kinematic transformations used to check for compliance with the tolerance can be outsourced to the commissioning of the mechanism. They are no longer required for the preparation of a specific motion. Only the motion itself has to be calculated once from the TCP to axis level. On the other hand, the axes are decoupled from each other by using $\lambda_{2.\text{corr.ax}}$ at axis level. This allows an optimized segmentation and a better adaptation of the splines to the motion specification.

For performance evaluation purposes, firstly the reduction of the number of calculated kinematic transformations is examined. As can be seen in Figure 14, this number has been reduced significantly for all algorithms. For the B-spline algorithms, 87.5% (bsLSQ$_\text{all}$, bsITER$_\text{all}$) to 95.8% (bsLSQ$_\text{max}$, bsITER$_\text{max}$) transformations are saved on average. This percentage is so high because the algorithms perform many transformations for searching the global maximum. For the algorithms that divide all segments (bsLSQ$_\text{all}$, bsITER$_\text{all}$), the saving is smaller since the computational effort itself is smaller. However,

since the kinematic transformations in the B-spline approaches only have a minimal impact on the total computation effort (Figure 8), no significant performance increase is expected. With the monomial spline algorithms, in average 63.8% (msITER) and 57.2% (msRECUR) of the kinematic transformations can be saved on average (66.7% and 58.7% for an accuracy of 1 μm). Due to the great influence of the transformations (Figure 8), this should be apparent in the overall performance.

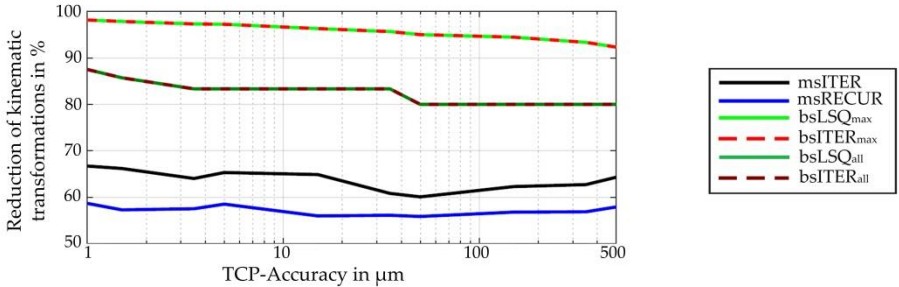

**Figure 14.** Reduction of the number of kinematic transformations by using the approach.

The next step is to consider how the accuracy of motion changes over the path. As described above, two factors lead to the calculation of a more accurate path than required. One is the reduction to a constant $\lambda_2$, which leaves up to 14% of the accuracy unused. The other is splitting $r_\varphi$ to the two axes via Equation (8). Here an error of up to 64% is expected. Figure 15 shows the deviation of the calculated spline from the optimized set points. (a) shows the conventional approach where all algorithms come very close to the allowed accuracy. In (b) the simple approach is shown. It can be seen that for the given motion a large part of the allowed accuracy is unused (>60%). Furthermore, it can be seen that for the algorithms that divide all segments (bsLSQ$_{all}$, bsITER$_{all}$) a much higher accuracy than necessary is achieved in wide ranges of the motion (e.g., between $\tau$ = 0.2–0.4). This reduces the performance of those approaches.

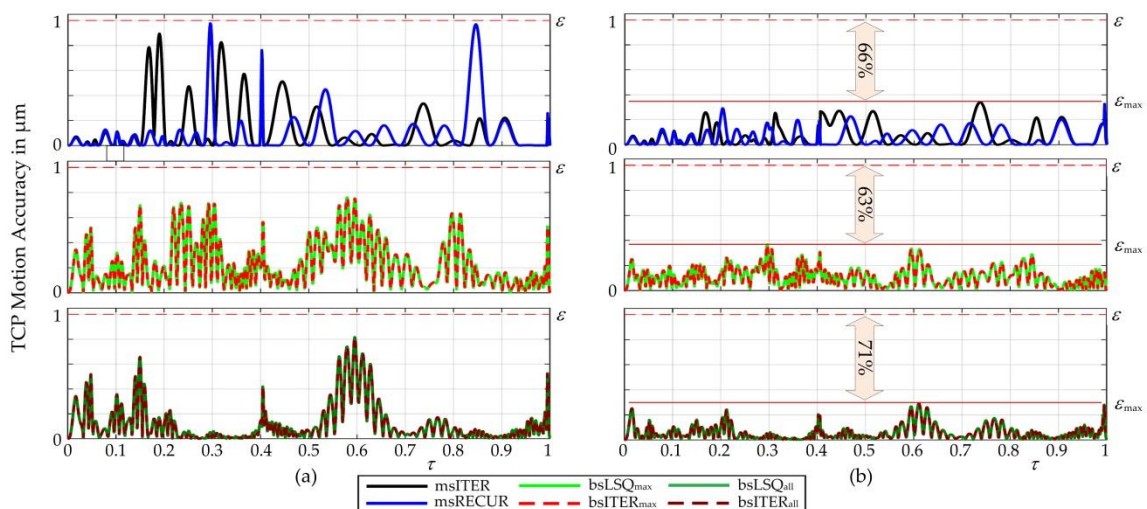

**Figure 15.** Motion accuracy of the calculated splines. (**a**) Using conventional an (**b**) simple motion preparation approach.

The results of the performance evaluation of the algorithms for decoupled axes are shown in Figure 16. The qualitative trend in (a) shows that the basic characteristic of the performance curves does not change compared to the coupled case. Due to the simplification made, the B-spline algorithms have a worse performance than before. At 1 μm accuracy, the number of generated data increases slightly by +13%, while the computational effort rises significantly by +30%. The higher computational

effort can be explained by the fact that the simplifications require a much higher accuracy than before. In combination with the great influence of algorithms on the total calculation effort, the effort increases significantly. As expected, the monomial spline algorithms generate more data than before, especially at higher accuracies. However, the overall computational effort decreases. This can be explained as the saved kinematic transformations have a high influence on the total computing effort.

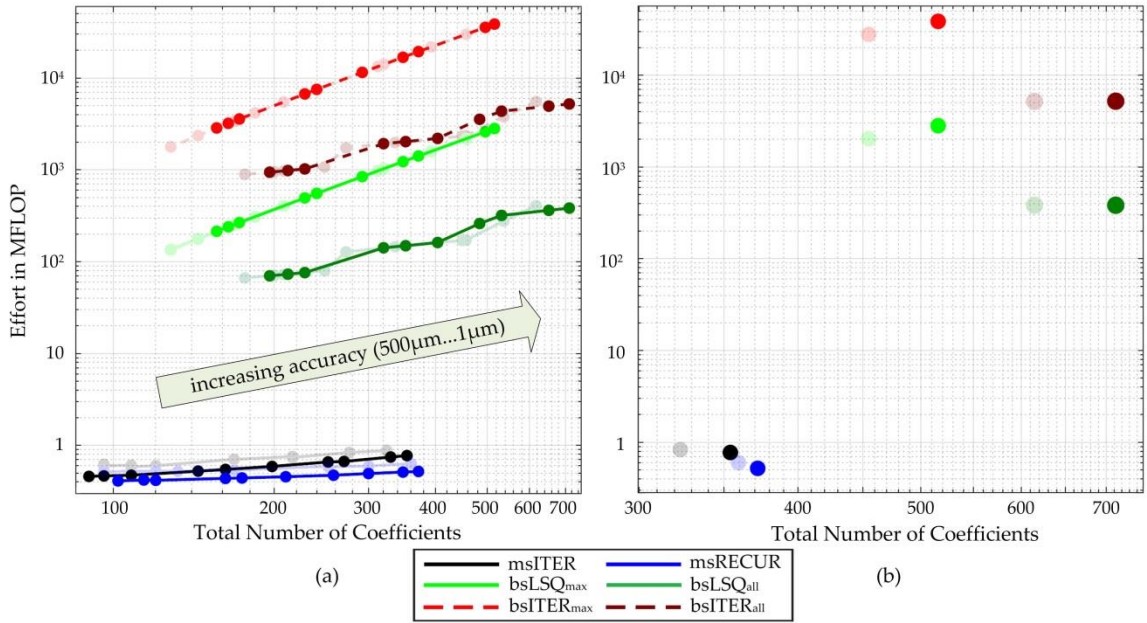

**Figure 16.** Comparison of the performance of the algorithms with decoupled drive coordinates. The values of the coupled calculation are shown transparent, cf. Figure 7. (**a**) Trend of the performance as a function of the accuracy. (**b**) Performance for the relevant accuracy of 1 µm.

For the example of 1 µm accuracy, msITER generates 4.8% less data than msRECUR. In contrast, msRECUR requires 32.8% less computing effort. In Figure 16b, msITER and msRECUR appear closer than in the conventional approach. Again, msRECUR is evaluated as the more suitable algorithm, since the savings in effort are much greater than in data. Once again msITER follows closely as the second best algorithm.

## 5. Discussion

### 5.1. Algorithms for Offline Motion Preparation

It could be shown on the example that the B-Spline algorithms investigated require a very high computational effort when they are used for offline motion processing. This is mainly because complex equation solvers have to be implemented in the algorithms. Due to these characteristics, the use of B-Splines cannot be recommended for offline motion preparation.

For a given accuracy of 1 µm, a recursive algorithm based on monomial splines (msRECUR) is evaluated as the most well performing. Compared to the iterative monomial spline algorithm (msITER), it saves up to 32.8% of the computing operations with only 4.8% more data.

Furthermore, it has to be summarized that as expected in all experiments msITER generates less data, while msRECUR requires the least calculation operations. If the amount of data calculated by the algorithm is more important, msITER is preferable. If the effort is more important, msRECUR should be chosen.

Through linearization of the non-linear mechanism, for the monomial spline algorithms the computational effort was reduced significantly by 19.1% with a minimal increase in the amount

of data by 3.3% ($\varepsilon = 1$ μm). This shows that even the simplest optimization approaches can have a considerable positive effect on performance of the algorithms.

## *5.2. Outlook*

Using the Jacobian matrix is an established method even in more complex kinematics [25]. Instead of ellipses, multidimensional ellipsoids with comparable properties then result, e.g., [29,30]. It is assumed that similar optimization approaches to those discussed in this work could be used even in more complex kinematics to minimize the computational effort. Additional investigations have to be undertaken to determine whether a transfer of the approach is possible or useful.

By adjusting the mechanism to the processes workspace, the non-linearities of the kinematics can be better represented by linearization. It is expected that this will further reduce the amount of required data.

In further work, ways should be investigated to exceed the assumption of a constant transmission in the entire workspace. A map-based approach seems to be useful here, similar to volumetric error compensation. With that, splines could be generated closer to the set points and the amount of data could be further reduced. In this context, the effects of grid spacing on discretization and estimation of a locally valid limit value should be investigated.

## 6. Conclusions

In this work, algorithms for offline motion preparation are evaluated with respect to their performance. On the one hand, the computational effort is presented. It determines the time required for motion preparation. On the other hand, the generated data volume is shown, because a small data volume is required in the machine's control system. The evaluation was carried out on the example of a non-linear kinematics and an example process from the field of processing machines. Characteristic for the process are comparatively large data volumes and high demands on the motion processing.

First, an obvious approach to generating motion splines from a given 2D point cloud at TCP level is presented. Selected algorithms based on B-Spline and monomial splines are examined for their performance. The evaluation is undertaken qualitatively by means of a 2D diagram, enhanced by the consideration of absolute values. After discussing the results, a recursive algorithm based on monomial splines is recommended for use in the example. Since the kinematic transformations have a negative influence on the performance, a simple approach for increasing the performance by reducing the total number of kinematic transformations is presented. With this simple approach, in the example of 1 μm accuracy the computing effort was reduced significantly while the data volume increased slightly. Higher reductions in computing effort are expected when transferring to more complex kinematics.

The evaluation showed that the monomial spline algorithms are much better suitable for offline motion preparation than B-spline based algorithms. Due to the large performance differences, a clear recommendation is given for monomial spline algorithms, even for other processes with similar requirements. For the example investigated, a recursive algorithm was identified as that which performed best.

**Author Contributions:** Conceptualization, O.H.; Data curation, L.D.; Supervision, S.I.; Validation, O.H. and L.D.; Writing—original draft, O.H.; Writing—review and editing, O.H., L.D. and S.I. All authors have read and agreed to the published version of the manuscript.

**Funding:** This research was funded by the German Research Foundation under grant number 182157057.

**Acknowledgments:** The authors would like to thank the German Research Foundation for supporting this work under grant number 182157057. Furthermore, we thank Christian Friedrich for his technical support.

**Conflicts of Interest:** The authors declare no conflict of interest.

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
