# Peer review of "Performance Evaluation of Offline Motion Preparation Approaches on the Example of a Non-Linear Kinematics"

_applsci, doi:10.3390/app10228014_

Round 1
Reviewer 1 Report
The paper presents a comparison of the performance of four different algorithms for offline motion preparation, in terms of required computational effort and data generated. A method to optimize the motion preparation process was presented, and its effects on the four considered algorithms compared. Several issues need to be amended:
- The scope of the considered algorithms in the paper seems very narrow. While the authors justify what spines were not evaluated (lines 176-181), the justification for what they do consider is poor. Moreover, phrases such as 'approaches that appear relevant' and 'approaches... consider important' should be revised to include better justification.
- The examples used to compare the algorithm performance in both Sections 3.3 and 4.2 need to be better described, so as to allow replication and verifiability. It is uncertain how intensive the examples used are, and how representative they are of a practical situation.
- Comparisons with percentages should be done more cautiously, especially since the change in actual numbers might be much larger than the percentages depict.
More minor issues with the paper are as follows:
- Most of the emphasis of the paper is on the motion preparation phase, with one method for optimization. The authors should clarify their explanation for Figure 2 more clearly.
- Line 68: A different word instead of 'Furthermore' should be used to indicate juxtaposition. Similarly, in line 112, 'on the other hand' indicates the next point contrasts the previous, even though the authors present both points as advantages.
- Several typos are present in the paper (line 92: approached; 207: inserted.s; 247: farer; 278: much higher (extreme is redundant); 313: therefor; 347: results to (in), 394: Jacobi matrix; 395: numeric)
- The term 'concrete manipulator' sounds like something used to handle concrete, as opposed to a practical manipulator.
Author Response
Dear reviewer,
we would like to thank you for your valuable feedback. The document attached will answer each point individually. Changed text passages are highlighted in blue in the new version of the manuscript.
Kind regards,
Olaf Holowenko

Reviewer 2 Report
- page 2 with reference to fig. 1-a and 1-c:
as explained in more detail in DOI: 10.1177 / 1687814018754953, the characteristic map of fig. 1-c is relative to speed profiles in the X direction (with reference to fig. 1-a), but the caption used here (Motion set point) could be misleading. I suggest to change it with 'V_x set-point' or 'Velocity set-point'. In fact, in the cited paper, the map of fig. 16 has much clearer axes captions.
I also stress that, as expressed in lines 105-106, the paper's work focuses on the map of position set-points. Wouldn't be better to provide displacement profiles as an example in fig.1-c?
- line 163 paragraph 3.1
The paragraph discusses the use of B-splines but it is not clearly expressed what type (univariate? Bivariate?). This is important because in case of univariate splines the nodal domain is presumably on a time scale, and the nodes associated with the points to be approximated are known (in fact it becomes the abscissa of the point to be approximated), if instead they must be calculated (in case: how it has be done?)
- line 199:
The methods based on B-splines (bsLSQ and bsITER) fall into the category of algorithms that Piegl and Tiller in the NURBS (cited in the article as a source [11]) define type 1 (i.e. they start from a relatively low number of control points and iteratively increase them to fall within the set tolerance). These methods are affected by a problem linked to the bisection method used (line 199) since when, during the insertion of the nodes, a nodal span is defined that does not contain one of the pre-calculated nodal values ​​corresponding to the points to be approximated return singular array in the LSQ method (see attached screen of NURBS, NURBS_typ1.PNG). In fact, in the more traditional LSQ methods these nodes are calculated with methodologies that guarantee a well-conditioned matrix (see attached screen of NURBS, NURBS_ubar_LSQ.PNG). This problem of convergence is not mentioned: how it has been managed? Moreover, the 'calculateEquidistantKnotvector ()' function used in both pseudo-codes suggests that in the first step the nodes have been calculated equidistant, a distribution that does not guarantee the condition mentioned before even in the first step (clearly this detail is not very influential since starting from a few control points it is unlikely that even with equidistant knots the first step is not respected).
- line 202:
The bsLSQ method is referred to [Engineering with Computers (2000) 16: 73–79 2000 Springer-Verlag London Limited), but in this source there is no mention of data approximation, much less of LSQ methods, but interpolation with constrained derivatives . This is also repeated in the pseudo-code of fig. 3.
- Figure 3 - 4:
* The structure of the pseudo-code suggests that at each iteration step only one node is inserted (in the center of the span where the maximum deviation is present) and then the calculation of the control points is re-performed with LSQ. This is not the most efficient method, in my opinion, to obtain the result. As reported in: https://www.researchgate.net/publication/24381198_Representation_of_Ice_Geometry_by_Parametric_Functions_Construction_of_Approximating_NURBS_Curves_and_Quantification_of_Ice_Roughness--Year_1_Approximating_NURBS_Construction
the recommended method is trivially to use some bool flags to mark the spans where the tolerance is exited, and at each cycle insert a node with the bisection in each of these spans. Clearly this way far fewer calculations than LSQ are performed and presumably almost nothing changes on the resulting curve (if at all) due to the local modification property. It would be necessary to address this point while explaining the results of Fig.7, even for what it said in lines 263,264,265.
- Figure 5 - 6:
Both a reference about algorithms used and an explanation are missing. How the system is defined (in matrix form with Vandermonde matrix?). It is not clear which algorithms are used.
- line 307 paragraph 4.1:
I find this part very interesting, as it evaluates how an approximation tolerance affects the displacement of the TCP (we could say in task space) on the joint space. In general it seems to me well explained.
- paragraph 4.2:
This part seems quite clear to me, showing diagrams of different indices with chromatic scale showing the entire workspace on the left and a hypothetical workspace on the right during use.
Author Response

(The authors gave the same response as above.)

Reviewer 3 Report
The paper presents an analysis regarding the performance of different motion preparation approaches known from technical literature. The performed research study evaluates some algorithms for offline motion preparation with respect to their performance, with applicability to a particular process. The paper conclusion highlights that the proposed recursive algorithm (based on Monomial splines) offers the best performance for the particular chosen example.
Manuscript’s strengths:
- The issue are well described and an overview about some solutions for these problems is presented.
Manuscript’s weaknesses:
- The research methodology regarding the proposed solution (linearization, Monomial splines) is not completely validated (too few tests performed for various other processes) to generalize the solution. The whole study focuses only on a particular case.
Recommendations for the improvement of the manuscript:
The paper relies heavily on bibliographic references 2-6 (Chapter 1. Motivation). For many untreated issues, references are made to these papers (optimization results, online aspects of the control).
A combination of three strategies is mentioned in this chapter, but it is not clear which are these strategies. Is the work focused on a specific one?
What means “conventional approaches” (compared to the proposed solution)?
Also, the paper focuses on one particular example. And all the conclusions refer only to this case. It would be useful to generalize the conclusions, at least for a certain class of processes (and not only for a particular one).
Many conclusions/results are “negative” (or vague). Therefore, in chapter 5 (Discussions) it say: “the B-spline-based algorithms are not suitable for offline motion preparation for the example discussed in this paper.” It should be specified (at least) for which class of processes/examples they are suitable.
I don't think numerical values are needed in the final conclusions. Analyzing the last sentence of the paper, it results that the proposed recursive algorithm (based on Monomial splines) is recommended for use in the example. So it can be used only for this considered case study?
The paper seems too focused only on one particular case. A rethinking of the paper for a more general solution (or generalizable conclusions) would be necessary. Otherwise, it seems only a study of a particular case, without the possibility of generalization and application to other processes.
Author Response

(The authors gave the same response as above.)

Round 2
Reviewer 1 Report
The authors have described the considered scope of the work in Section 2 in relation to processes similar to those in Section 1, which greatly clarifies the paper. The content of the manuscript is also easier to follow now. Some very minor remarks:
- (Line 73) Consider using 'A disadvantage' instead of 'Disadvantageous', which is an adjective.
- (Line 252) Please also state that bsITER_{max} and bsITER_{all} consider the same two modes as those for bsLSQ_{max} and bsLSQ_{all} for clarity.
- (Line 311) 'distance' should be 'difference'.
Author Response
Dear reviewer,
many thanks for the positive feedback. Your latest suggestions for improvement were considered in lines 73, 223f, 254f, and 314. All your remarks have helped us to improve the quality of the manuscript.
Kind regards,
Olaf Holowenko
Reviewer 3 Report
After the revision, the paper was significantly improved and many aspects are clarified.
The authors responded to all requests and comments in the revised form of the manuscript, clarifying the reported aspects. Two additional variants for the use of the B-Spline algorithms were added. The main request, a generalized conclusion, was also accomplished. Overall, the paper is well written and the issues are well described.
Author Response
Dear reviewer,
many thanks for the positive feedback. Your remarks helped us to improve the quality of the manuscript.
Kind regards,
Olaf Holowenko